# A qualitative exploration of young people's experiences of attempted suicide in the context of alcohol and substance use

Rebecca Guest[1]*, Alex Copello[1,2], Maria Michail[3]

1 School of Psychology, University of Birmingham, Birmingham, United Kingdom, 2 Research and Innovation, Birmingham and Solihull Mental Health NHS Foundation Trust, Birmingham, United Kingdom, 3 Institute for Mental Health, School of Psychology, University of Birmingham, Birmingham, United Kingdom

☯ These authors contributed equally to this work.
* Rebecca.guest@nhs.net

**Data Availability Statement:** All relevant data are within the paper and its Supporting information files.

## Abstract

The aim of this study was to explore young people's experiences of the role and the processes underpinning the use of alcohol and/or other substances in attempts to end their life. Seven young people, aged 16–25 years old, were interviewed using in-depth, semi-structured interviews. Interpretative Phenomenological Analysis was used to analyse these interviews and develop an understanding of how young people understand their attempted suicide in the context of alcohol and/or other substance use. The analysis identified four superordinate themes reflecting young people's experiences across the seven interviews. Superordinate themes included: i) The complexity of relationships; ii) The double-edged sword of alcohol and substance use; iii) The straw that broke the camel's back; and iv) Reflecting on the on-going processes of recovery. The results of this study highlight the complex and multifaceted functions of the consumption of alcohol, and other drugs, in the experiences of young people attempting suicide. Young people described a number of inter and intrapersonal factors which impact upon their suicidal experiences including suicidal ideation and attempts. Participants reported using alcohol and substances as methods of coping with distress, low mood, hearing voices, anxiety and mania. However they also reflected on the impact that this has on their own suicidal ideation and attempts.

## Introduction

### Suicide and young people

Suicide is the second leading cause of death in 15 to 29-year olds globally [1] with evidence pointing towards a sustained increase in recent years—greater than in any other age group [2]. Suicide is complex and often has more than one precipitant, including adverse childhood experiences (ACEs), abuse, neglect, academic pressures, substance and alcohol use; and, mental and physical ill health [3]. The most significant predictors of completed suicide include a prior suicide attempt and previous self-harm, defined as any act of self-injury carried out by a

**Funding:** The author(s) received no specific funding for this work.

**Competing interests:** The authors have declared that no competing interests exist.

person, irrespective of their motivation [4]. Fifty per cent (50%) of young people who die by suicide have previously self-harmed [5]. Perceived family support [6] and a sense of meaning in life [7] are cited as being protective factors from suicide among young people.

## The role of alcohol and substance use in youth suicide

The links between alcohol, substance use and suicide are well documented [8–11] as are the mechanisms underpinning this relationship including impulsivity, depression and hopelessness [12]. Evidence from the literature point towards a vicious cycle of individuals using alcohol to cope with the distress associated with suicidal experiences, for example suicide ideation which in turn can increase the prevalence and distress attached to those experiences [13]. Esang & Ahmed (2018) [14]reported that drinking alcohol at a young age, alongside binge drinking leads to increased suicide ideation in adulthood. They explain that those with alcohol dependence or substance use are more than 10 times more likely than the general population to die by suicide.. Indeed, evidence from the National Confidential Inquiry [5] found of those 20-24-year olds that died from suicide 40% had excessive alcohol use and that 33% of under 20s who self-harmed had high rates of excessive alcohol and 42% used substances. Although the association between alcohol, substance use and suicidal behaviour seems well-established [15] (Borges et al, 2016), we do not know how young people understand and experience this relationship; and if and how they see the role of alcohol and substance use as an important contributing factor to their suicidal behaviour. Witt and Lubham (2018) [16] have argued that, those who use alcohol or substances are frequently excluded from taking part in suicide prevention studies. Overall, there is insufficient attention paid to alcohol and other drug use in suicide prevention strategies.

The views of people with lived experience, including young people, should be at the heart of suicide prevention policy, guiding services on how to support these individuals in a way that is feasible and acceptable to them [3]. It is therefore vital to gain a deeper understanding of young people's experiences of the role of and processes underpinning the use of alcohol and/ or substance in attempted suicide. This could help us improve the identification and management of alcohol use as a suicide risk factor among young people, thus, informing future suicide prevention interventions for this population. For the purpose of this study, the term 'substance use' is used to describe any illicit drug use, not including alcohol.

## Materials and methods

### Design

A qualitative study based on semi-structured interviews with young people aged 16 to 25 years old who were under the care of a youth mental health service. The study was conducted in line with the consolidated criteria for reporting qualitative research [17] and the completed checklist can be found in S1 Appendix.

### Material

The interview topic guide was developed in collaboration with young people from a Patient Advisory Group from a local National Health Service (NHS) trust to ensure the questions were meaningful to the end users of the research; that the language and wording was acceptable so as to reduce potential negative impact on the participants. Members of the Patient Advisory Group were within the target age range of the study sample and all had their own lived experience of mental health difficulties including suicidal experiences. The group was also consulted on the acceptability of the participant information sheet, consent form and debrief sheet.

Additionally, the group helped to identify which services might be helpful for young people to know about following their participation in the study.

Ethical approval was obtained from the Health Research Authority (HRA). Research Ethics Committee reference: 19/WM/0082. Integrated Research Approval System: 257738.

### Participant recruitment

A purposive sampling method was utilised in order to identify participants for the study.

Participants were recruited from two NHS trusts in the UK who provide mental health services to individuals aged 16 to 25 years old. The first author (RG) attended multi-disciplinary team meetings to speak to team members about the research; and, requested staff to identify any individuals who met the study criteria, provide them with information about the study and gain consent from the young person for RG to contact them directly to discuss their participation in the study. See S2–S5 Appendices for recruitment information. Potential participants were contacted via email or telephone to ensure they met the inclusion criteria and answer any questions they had about the research before agreeing a time and date for the interview.

Inclusion criteria were as follows:

- Aged 16–25 years old

- Be under the care of the local Child and Adolescent Mental Health Service (CAMHS)

- Have a self-defined 'problematic' use of alcohol (and substances). A measure of alcohol use was deemed inappropriate by the young people's group that was consulted during the development of the project. Also, what is 'problematic' drinking is idiosyncratic.

- Speak fluent English

The only exclusion criterion involved having made a suicide attempt in the six months prior to taking part in the project due to attempts to limit the impact of participation on the individuals and manage risk, in line with advice received by the Research Ethics Committee.

### Data collection

Seven semi-structured interviews were conducted by the first author (RG) with each participant interview, ranging from approximately 40 to 75 minutes in length. All interviews were audio-recorded and later transcribed verbatim. Five interviews were conducted at the local mental health team where participants were usually seen by the teams that supported them. The other two interviews were conducted over the telephone at the participants' request. The topic guide questions covered: young people's experiences of attempting suicide, how things had changed, if at all, since their last attempt and what their experiences of alcohol and substances were and how, if at all, they thought the use of alcohol and substances affected their mental health. Following the interview, participants were offered an optional debrief session and were given a debrief information sheet with information of support services, none of the participants requested a debrief with a Psychologist that was independent of the study. Participants were given two weeks to withdraw from the study. None of the participants requested that their data be removed from the study.

### Data analysis

Each transcript was analysed by lead researcher (RG) using the Interpretative Phenomenological Analysis (IPA) process, as described by Smith, Flowers & Larkin [18]. IPA was chosen

given the focus on the participant's individual and shared personal experiences rather than pre-defined categories or developing an overarching theoretical explanation. IPA enabled detailed exploration of how young people made sense of their experiences and therefore a total of seven participants was deemed acceptable [18]

The first step of analysis involved 'reading and re-reading' each transcript in order to become familiar with the data. The next step involved 'coding' focusing on descriptive, linguistic and conceptual comments. After coding each transcript, potential identified subordinate themes were written on pieces of paper and arranged into possible overarching superordinate themes. Following this, a table of 'emergent themes' was created which comprised of an outline of superordinate and subordinate themes including quotes. A short reflections sections was also added on the experience of the interview and the analysis process. This process was followed for each participant. The emergent theme tables were then sorted using a similar paper cuttings exercise to identify any recurring, shared or contrasting themes across individual participants and were again sorted into superordinate and subordinate themes to develop a final thematic structure. Throughout all staged of analysis interpretations and ideas were discussed with a qualitative research peer group and supervisors in order to ensure the process was as robust as possible. At the time of the research RG was a female Trainee Clinical Psychologist, this was considered when reflecting on any findings.

## Sample

Six of the participants identified as female and one participant identified as male. Ages ranged from 16 to 24 years old (mean = 20) and all reported ethnicity as White British. All participants identified at least one occasion where their alcohol use had been 'problematic' and all spoke about using cannabis, with some also discussing their use of other substances e.g. methylene-dioxymethamphetamine (MDMA), ketamine and cocaine. All participants were under the care of their local CAMHS. Some participants described on-going experiences of suicidal ideation and more recent experiences of attempting to end their life. Other participants described their mental health as 'stable' and described previous historical attempts. Table 1 below provides brief information about each participant. All participants were given a pseudonym.

## Results

During the interviews, a number of important and powerful experiences were discussed related to interpersonal relationships, using alcohol and substances, harm to self and recovery. The analysis identified four superordinate themes and twelve subordinate themes which demonstrated how participants made sense of their experiences of attempting suicide and their understanding of the role of alcohol consumption and substances in relation to their mental health. This thematic structure is described in Table 2 below. See S6 Appendix for further supporting quotes.

### 1. The complexity of relationships

This superordinate theme was discussed by all seven participants. Participants described difficult relationships with friends and family some of which have triggered or contributed to their mental health difficulties. However, they also spoke about the importance of feeling supported and able to talk to others.

**1.1 Keeping safe from others.** All participants described experiences of other people being unhelpful, not understanding their mental health needs and being treated in a stigmatising way by others. Participants often expressed feelings of not fitting in with others and a need to be independent and not a 'burden' on others. Some participants had experience of others

**Table 1. Information about the research participants.**

| Pseudonym | Demographics and other information |
|---|---|
| 1- Annie | Annie currently lives with her parents and one sibling. She reported on-going suicidal ideation and a number of difficult current life experiences and circumstances that were impacting on her mental health. She explained that she is still accessing support from CAMHS and a Psychologist. She described attempting to end her life on more than one occasion and talked about how her use of solvents was closely linked to these attempts. Annie also had used alcohol and cannabis as a coping strategy and would still occasionally drink alcohol or smoke cannabis. Annie appeared to become angry as the interview progressed and spoke in detail about her experiences. |
| 2- Liz | Liz currently lives with her boyfriend who attended the interview with her. She reported that she was still accessing support through CAMHS and had previously engaged in Dialectical Behavioural Therapy (DBT), however was due to be discharged from the service and described her mental health as 'stable'. Liz reported attempting to end her life on more than one occasion however appeared uncomfortable at times going into further detail about some of her experiences. Her boyfriend reported at the end of the interview that there were two main life events that impacted on Liz's attempts to end her life; however, she did not wish to discuss these. Liz spoke about her use of alcohol, cannabis and the misuse of prescribed medications. |
| 3- Sarah | Sarah currently lives with her boyfriend. She reported that she is still accessing CAMHS and had previously engaged in DBT. She described that her mental health is currently 'stable' and was open and reflective throughout the interview. Sarah spoke in detail about her use of alcohol, which she now limits, and the impact of this on her mental health. She also described occasionally smoking cannabis. Sarah spoke about previous self-harm and overdose attempts which appeared to increase in severity prior to her being supported by CAMHS. |
| 4- Matt | Matt is the only male participant. He currently lives with his dad and stepmother. Matt was the only participant who described only using alcohol problematically on one occasion, as part of a suicide attempt. Matt also described the use of cannabis and its impact on his mental health. Due to the alcohol use and overdose, Matt has patchy memories of trying to end his life. He talked about his negative experiences of services and the impact of this on his mental health. |
| 5- Louise | Louise currently lives with her parents. She reported that she is still using drugs and still has times when she struggles with her mental health and suicidal ideation. She is accessing support from CAMHS and is engaging in DBT as well as having support from a substance misuse service. Louise spoke about a number of attempts to end her life, using a variety of methods. Louise drew strong links to substance and alcohol use, her mental health and suicidality/ self-harm. |
| 6- Holly | Holly lives with her mum and spoke about the importance of her mum's support. She described her experiences of attempting to end her life and use of alcohol. She also described occasional use of cannabis. She spoke about her experiences of services, being given different labels, the use of mental health terminology and her attempts to make sense of this. This all appeared to have an impact on her mental health and identity. |
| 7- Belle | Belle lives with her parents. She explained that she feels her mental health is 'stable' most of the time. Belle spoke in detail about her first attempt to end her life and discussed other attempts and methods that she had thought about. Belle spoke in detail about the impact of drug and alcohol use on her mental health. Belle explained that she is currently still using drugs and alcohol. She described the importance of her relationship with family and friends. |

being harmful, for example, being a victim of domestic violence, sexual assault or witnessing parental or gang related conflict. These experiences were then related to the development or perpetuation of mental health difficulties and suicidal ideation. Annie described her experiences of other people being violent towards her:

> "…. got a boyfriend, got into an abusive relationship, he used to beat me, he said he had an attachment disorder, if I would refuse to go he would tell me to kill myself, slit my throat, slit my wrists. So I would go and then get beat up and now it's my little brother beating me up and saying nasty shit"
>
> (Annie)

It appears that having such experiences would consequently impact on the individual's willingness to engage with and trust the intentions of others.

**Table 2. Participant themes.**

| Superordinate themes | Subordinate themes | Participants that contributed and % |
|---|---|---|
| 1. The complexity of relationships | 1.1 Keeping safe from others | 7/7 (100%) |
| | 1.2 Still needing a connection despite the difficulties | 7/7 (100%) |
| 2. The double-edged sword of alcohol and substances | 2.1 Using alcohol/ substances to escape unwanted emotions | 7/7 (100%) |
| | 2.2 The adverse impact of alcohol/ substances on mental health | All (100%) |
| | 2.3 Others encouraging and normalising use | 5/7 (71.4%) |
| | 2.4 The changing and unpredictable use | 6/7 (85.7%) |
| 3. The straw that broke the camel's back | 3.1 A gradual build up | 5/7 (71.4%) |
| | 3.2 Being determined to harm the self | 5/7 (71.4%) |
| 4. Reflecting on the on-going process of recovery | 4.1 Increasing understanding of the self and experiences | 5/7 (71.4) |
| | 4.2 Using alternative coping strategies | 7/7 (100%) |
| | 4.3 Taking responsibility for recovery | 5/7 (71.4%) |
| | 4.4 The emotional difficulty of reflecting | 6/7 (85.7%) |

Each theme is described in detail below with illustrative verbatim quotes from participants included in order to support these descriptions.

Belle also spoke about how others' lack of understanding and stigmatising views were difficult to negotiate when trying to explain her experiences:

*". . ..so for me, to come in and try and explain to my parents who thought the people that self-harmed were seeking attention, to try and go and explain that to them, they were really confused they didn't have a clue what was going on, it's quiet funny looking back at it actually, urm you know, they didn't have a scooby what was going on do you know what I mean"*

*(Belle)*

This is an experience echoed by other participants who found that the "ignorance" of others regarding awareness of mental health problems was difficult for them in trying to gain support and help family and friends to understand. Another complication was a described 'mis-match' in understanding *"I mean I still wanted to go (end his life), I think it was more, it was like a, I mean I don't see it as a cry for help but a lot of people do"*(Matt).

Many of the participants spoke about a feeling of not fitting in with others due to experiences of bullying, being excluded and mental health professionals struggling to find explanations for their reported symptoms. This notion of not fitting in then impacted upon their mental health and self-worth.

*"I hated school, I didn't really get on with anybody urm, I just kind of was by myself and when I was in a group, I would always try and be like urm, the silly one, who always tries to make everyone laugh, just to try and fit in [ok] but it didn't work so I would I would get, the voices in my head would tell me like oh you're worthless and things so I'd feel suicidal at school"*

*(Louise).*

These experiences of individuals not feeling heard, cared for or actively harmed culminate in further feelings of isolation and threat which are known precipitants for mental distress [19, 20].

**1.2 Still needing a connection despite the difficulties.** Despite the reported difficulty in relating to, and connecting with, others as described in the first subtheme, all participants described the importance and value in feeling supported, understood and heard by family,

friends and professionals. This tension between difficulties connecting yet needing connections was evident from their descriptions. They explained how difficult it is to talk to another person about their mental health, especially related to suicide, but also how helpful this can be. Liz talked about how being surrounded by supportive and positive individuals had a beneficial influence on her mental health:

> "yep. . .and I'm with happy people. . .personal support (mumbles), I've got his family, my family, I've got all these people round me and I've got some good friends as well, whereas before, I had some really toxic people"
>
> (Liz)

In addition to feeling supported by others, participants spoke about the importance of being able to talk to people about their feelings and experiences. Sarah explained that when it is difficult to talk to others, it can be helpful for people to ask how you are:

> "I think we just need to check in on people, you know, if people genuinely sat me down and asked me if I was doing ok, I would have been honest, but nobody asked, you know I am not blaming anyone else or saying that it's their fault but you know small things like that can really make a difference and if not stop, but at least delay something happening"
>
> (Sarah)

Belle also expressed how challenging it can be to reach out; nevertheless, stressed how important she feels that talking to others is and credited talking as one of the reasons that she is here to today.

> "I think I can, the one thing I can say is, just when someone is feeling like shit, just talk, that's all you need, you need to talk out do you know what I mean and I'd like you to make that very obvious because if I hadn't talk out I wouldn't be here, I wouldn't be alive"
>
> (Belle)

> She *they're all involved in mental health some*

It appears that difficult life experiences and relationships with others, as well as feelings of not being understood and excluded by others is significant in making sense of participants' experience of attempting to end their life. Whilst supportive relationships with others, family, friends or services can provide hope and containment in very difficult times and for some individuals they felt that this was vital in them still being alive, at other times these same relationships could be very challenging.

## 2. The double-edged sword of alcohol and substances

This superordinate theme was discussed by all participants. Individuals spoke about using alcohol and substances to help them manage their overwhelming emotions in response to difficult life events, their mental health difficulties and other stressors. They also talked about the impact of this use on their mental health, suicidal ideation and actions. Some participants spoke about the influence of others on their use of alcohol and substances and how their use may have changed over time.

**2.1 Using alcohol and substances to escape unwanted emotions.** All participants shared experiences of using alcohol and substances to cope with overwhelming emotions including significant anxiety and low mood. Although, for some the use of alcohol and substances served a different purpose, for example; *"probably both, I used to drink and smoke at the same time, it was more the cannabis that made me feel relaxed the alcohol used to make me feel really hyper [laughs]" (Liz).*

As the quote illustrates, Liz used alcohol in order to improve her mood and cannabis to calm her down. Sarah spoke about using cannabis to cope with 'feeling risky' or with experiencing suicidal ideation, comparing cannabis to pro re nata (PRN, when necessary) medication:

*"I find it's (cannabis) kind of a. . . (long pause) almost like a PRN in the sense of you know, if I find myself feeling a little risky it kind of it just, it lowers my risk, I'm not able to. . .I can't think if what I want to say sorry. . .I think it's just kind of the lack of thinking, like I can't think, I can move but it's not, you know it's that kind of, I physically am unable to hurt myself if I wanted to, its more effort"*

*(Sarah)*

Holly also spoke about using alcohol to help down-regulate her system and how it became a coping strategy to aid in managing her 'high energy':

*"I just noticed the effect, like being a student you drink right so, and then you learn about how alcohol affects you then, and if I calms you down, when you want to be calm you be like oh I'll just have a drink [ok] I realise that's not healthy but at the time I didn't care so [ok]"*

*(Holly)*

Other individuals spoke about how alcohol and substances were both used in response to interpersonal conflict and the difficult emotions triggered as a result of this. Louise talked about using substances as a way of numbing the emotional pain triggered by family problems:

*"well I was going through a bit of a rough patch, in my life where I found out that my dad wasn't my real dad and stuff [right ok], erm and it just kinda, I didn't really care anymore. So I was just like, I'm gunna start taking drugs and its helps and heals the pain by (mumbles) but in the end it doesn't, it doesn't actually help with the pain [right] it just make things worse"*

*(Louise)*

Belle, on the other hand, spoke about alcohol as more of an automatic 'go-to' coping strategy after a stressful interaction with a significant other. This suggests that alcohol had become a learnt way of coping for her at that time:

*"as soon as like that happened with him, I just went out drinking and I got drunk every night it was awful and I was just in such a bad way I just wanted to forget about it urm it made me feel worthless, I made me feel like shit".*

*(Belle)*

Alcohol and other substances, particularly cannabis, were used as coping strategies in response to what participants described as having a 'bad day', interpersonal conflict or

overwhelming emotions. For some individuals, different substances served different purposes, and some felt more helpful than others. For example, cannabis was used by young people to reduce anxiety and alcohol in an attempt to improve mood.

**2.2 The adverse impact of alcohol and substances on suicidal experiences and mental health.** All participants reflected on the negative consequences of using alcohol and other substances as a way of managing emotions and interpersonal conflict. This subtheme highlights the effect coping in this way had on their suicidal ideation and other comorbid mental health difficulties. Annie explained that, in her experience, substances such as solvents and cannabis had negative effects on both her physical and mental health, whereas alcohol appeared to be less of a problem and potentially seen as helpful:

*"I didn't eat for pretty much that whole two years, I barely ate, I ended up getting, I've got an eating disorder now, so it's not only that it's probably, its ruined my insides, I mean, after I stopped doing solvent abuse, I threw up every day for about 3 months after it, I mean it's just. . ..alcohol, if you monitor it, it can help you, I don't knw about weed that much because it sent me paranoid and it messed with my anxiety but solvent abuse, it fucks you up, that's the best way I can say it."*

Louise described a somewhat different experience of alcohol use, making clear links between consuming alcohol and an increase in her suicidal ideation and behaviours:

*"when I drink erm, I get, obviously I get drunk [yeah] and the normal symptoms and that and then urm, if I continue to drink after that, I go insane, like completely insane, well the once I did it, I went in the shower and I started to self-harm with a razor [ok] and my mates were there and they kicked down the door and took me out the shower and then I ran off and tried to jump out of my window to end my life urm, so that was one experience I've had, urm, pretty much all similar on drink I just always feel suicidal. . . always get memories from the past [ok] and urm it just brings everything up and I just, I can't deal with it so I end up self-harming, I end up trying to take my life or having thoughts of taking my life"*

*(Louise)*

Although not all young people described such a concrete link between alcohol and suicide, they mostly identified that alcohol did have an impact on how they felt the day after drinking. They spoke about becoming more emotional and less able to function, which in turn led to increased levels of distress.

Participants described a range of different ways in which alcohol consumption affected their mental and physical health and relationships. All participants suggested that there is an effect that may at the very least contribute to an increase in distress. This distress can then become part of an experience which leads to increased suicidal ideation. Others discussed a direct link between the use of alcohol and substances and an increase in thoughts to end their life.

**2.3 Others encouraging and normalising use.** Five participants contributed to this subordinate theme. They described experiences of other people introducing them to alcohol or substances or downplaying their use. Belle reported being introduced to using drugs by an ex-boyfriend and that this led to her using a number of different substances. Holly described that *"my dad was addicted to alcohol so I guess that was another, oh this is a coping mechanism not thinking for myself, this isn't healthy (laughs) . . .yeah drinking, smoking, drugs [ok] anything that would mind alter he did it"*. Sarah also described experiences of family members

excessively drinking alcohol and therefore her drinking was minimised and not taken seriously. She reported that society also normalises alcohol use:

> *"you knows it's, even when my drinking was at its worst it was still seen as, oh she enjoys a glass of wine, people still thought it was this, funny, social, oh she just likes getting drunk, where as in reality it was a way to destroy myself, and I think people see it as something fun and light and it's not"*
>
> *(Sarah)*

Liz and Matt spoke about using substances in more of a social context with peers; *"I guess it's just socialising, if he rolls up a spliff or whatever [yeah] and we just end up smoking it" (Matt)*. All of these participants appear to associate some of their alcohol or drug use to others and the influence that they have.

**2.4 The changing and unpredictable use.** Individuals also described how their use of alcohol and substances has changed or is continuing to change in response to reflections of the impact that this has on them; and, also the unpredictability of the effect of alcohol and substances on their mental health and suicidal experiences. Annie and Holly both reflected on being aware of their 'limit' and how alcohol may affect them; *"I don't get obliterated, cuz I don't know whether I'm going to be a happy drunk or an angry drunk, I don't know, but I don't push myself to that limit with alcohol" (Annie)*. Sarah and Louise also commented on the unpredictability of the effect of drugs and alcohol:

> *"it just kinda got me out my head, sometimes, you know I'd get really giggly and I'd watch comedy shows, like have a laugh, (quietly) the other times it would be really bad. There was no, kind of in between, by the end of the night I was either. . .high with joy or completely just depressed, there was nothing in between."*
>
> *(Sarah)*

For some participants, this unpredictability seemed to be a deterrent to using alcohol or drugs, however for others the benefits of using appeared to outweigh any potential negative consequences. Matt spoke about how his use of alcohol changed during his suicide attempt, and that this was unusual for him as he did not see himself as someone who would use alcohol as a coping strategy. This highlights how young people can use alcohol and/or substances in ways that are unpredictable or different to what they used to, especially in times of crisis:

> *"it's never been something that I've gone to when I'm depressed, I've got the self-harming techniques that I grew up with I guess so I guess I dunno drink is just not one of them. Obviously, I had the prosecco when I took my overdose"*
>
> *(Matt)*

## 3. The 'straw that broke the camel's back'

This superordinate theme is a direct quote taken from Sarah's interview when she talks about the gradual build-up of events and final trigger before she attempted to end her life. All participants acknowledged that there were contributing factors in place which led up to their attempt on their life. Although their respective experiences are quite different, some described an impulsive act, whereas other described researching and planning their attempt.

**3.1 The gradual build-up.**   Some participants described the involvement of other people in the build-up prior to their attempt. Annie explained that: *"people push you to do it, or things push you to do it, do you know what I mean, there's only so far you can stretch an elastic band before it snaps"*. Sarah also described experiencing bullying and a breakdown of a relationship as triggers to her attempt. There was often more than one event, in different areas of their life that may have happened at similar times:

*"I mean there was a few things that happened at the time, urr, I had a girlfriend, I was doing, well I was over working myself, I was doing about 80 hours at work [oh gosh] urr and I was going to college as well (laughs), so yeah, urr, it was just everything coming to an end really, cus I was working so much, I lost my college placement [right] and then I got over tired and I got urr, well I had to had in my notice cus of things that were happening, and my relationship ended so. . .urr it was just everything kind of coming crashing down"*

*(Matt)*

Matt explains how a combination of the end of his relationship, work pressures, college pressures and lack of sleep all contributed to his attempt to end his life. For others, the build-up was not be so clear, especially if their attempt related to a deterioration in their mental health, which may have impacted on their sense making at that time. However, lack of sleep is also something highlighted by Holly as an influence in her alcohol use and ultimately her suicide attempt. Holly explained that times when she has self-harmed have generally been linked to periods of 'hypomania' or *"energy swings"*. She described that before a suicide attempt, she had not slept for 6 days and was consuming alcohol to try to help her sleep.

**3.2 Being determined to harm the self.**   Many participants described a sense of determination in relation to ending their life at the time they made their suicide attempt. They explained that they did not feel as though anything could have been done to prevent them from taking the actions that they did. However, some individuals explained that their attempts were impulsive; this appears to have been facilitated by the consumption of alcohol or substances at times. Others planned and researched the ways in which they would try to end their life, both with and without the use of alcohol or substances. Belle, Annie and Louise all described spending time thinking about methods of harming themselves:

*". . ..then there was other attempts like with the bleach and I was going to take an overdose again, again I was planning to hang myself or I used to plan a lot, falling down the stairs, trying to, which I know sounds really stupid, but you know, you'd really injury yourself more than dying falling down the stairs but, I used to think, if I get the right trip, I can die and I remember I used to google how to break your hand and stuff like that like to try and hurt myself, urm and I used to jump off the bed to try and hurt myself, to try and break an arm or really really harm myself, where I could be in a cast for like 6 weeks you know, I was trying to do stuff like, and urm you know the thought of jumping of a bridge or anything, I'd be petrified because I am petrified of heights so I would never do that and I never thought about doing it either, I think mainly it was taking something that I thought would be the best way"*

*(Belle)*

Several other participants spoke of the feeling of inevitability of making an attempt to end their life, *"honestly, I don't think so, I think. . .it was such a dark place to be in that nothing, or at least it felt like nothing in that moment could have pulled me out, nothing could have stopped it"* *(Sarah)*. This determination to harm the self was described as ruminating about ways in which

to do this. Annie talked about how she would find another method to use if her access to her preferred option was reduced. However, Matt did not seem to share this experience and instead described a 'moment of madness' and being in a process of still making sense of what led up to him taking the actions that he did.

## 4. Reflecting on the on-going process of recovery

This superordinate theme considers the process of recovery and how this was talked about and made sense of by the participants in their interviews. Participants discussed how coping strategies (such as using alcohol/substances) changed alongside a developing understanding of their experiences. This sense-making of their experiences of attempting to end their life appears vital in their ability to reflect and move away from self-harming.

**4.1 Increasing understanding of the self and experiences.** Participants reported a number of factors which helped them to increase their understanding of their experiences to end their life and their mental health more generally. Often this increase in understanding was related to support given by mental health services or other professional support. Holly and Belle both identified that the process of receiving a diagnosis that they felt provided a helpful explanation of their experiences was difficult. However, once they received a label that they related to this helped them to make sense of their experiences and provided them with some explanation, although this was a process in itself;

> *"I think it is very representative, I think it is very representative of a lot of people, a lot of young women, to get political urm who have a diagnosis of borderline personality disorder, personally I disagree when the young person is going through puberty, the emerging bit exists for a reason but I think that has a big factor because I was diagnosed with BPD, EUPD whatever you want to call it and then, when I went home and did my research I was like, what the fuck are you on about? This isn't me, what, I don't know what you're on about urm, and then when I got the bipolar diagnosis initially, I was like oh this answers everything oh my god"*

> *(Holly)*

Louise also spoke about the influences of educational support in understanding the impact alcohol has on her: *"yes they (local drug and alcohol service)have, that's another reason why I've kind of stopped alcohol because they've you know told me about it and I've just realised that alcohol doesn't help at all"*. She spoke about the links she had been supported to make between her increase in suicidal ideation and actions after she consumes alcohol by a local drug and alcohol misuse service. Other participants talked about engaging in Dialectical Behavioural Therapy (DBT) and how this facilitated their understanding of their experiences. Matt described:

> *"I think that's what DBT has helped me with ur, especially working with (clinician name) urm I guess being in that 1:1 scenario seeing the difference of how its affected me, talking about it, I think before I just got overly anxious and just really kind of annoyed and ended up kind of zoning out and just getting too annoyed to speak about it [mm] but I think now being able to, I mean especially with (clinician name) I was able to speak about like, especially the bad experiences I've had, and it's just like being able to accept how angry I got at the time and how it didn't help"*

> *(Matt)*

Increased self-awareness and understanding appears to support individuals in the recovery from suicide attempts and behaviours that may have perpetuated these experiences, such as self-harm and the consumption of alcohol and substances.

**4.2 Using alternative coping strategies.** All participants discussed the use of alternative coping strategies to self-harm and the use of alcohol or substances as part of their recovery process. These varied from skills that they had learnt from engaging in DBT, to distraction techniques, exercise and self-care: *"I go to the gym a lot, and I go for walks and I talk to people and I don't delve into impulses like before" (Holly)*. The strategies described appeared to be effective in reducing harm to self and alcohol or substance use. Other DBT skills discussed were mindfulness and interpersonal effectiveness strategies. These appeared to help individuals reduce feelings of anxiety and increase their sense of being in control of emotions.

Annie, on the other hand, spoke about alternative coping strategies which seemed to extend the repertoire of options available to her when she was feeling overwhelmed and experiencing suicidal thoughts:

*"I started smoking cigarettes and vape, urm I started smoking cigarettes when I was like 11, urm and that was like a coping mechanism and then I ended up stopping when I went into year 10, didn't work and then I got a vape and I use my vape now, but apart from that I've got four cats, that I just try and surround myself with them"*

*(Annie)*

These alternative strategies seem to extend the repertoire of options available to these individuals when they are feeling overwhelmed and notice an increase in suicidal ideation.

**4.3 Taking responsibility for recovery.** Individuals spoke about taking responsibility for their recovery from self-harm and suicide attempts but also from their drug and alcohol use. Belle spoke about the importance of being in the 'right mindset' in order to get better and making sure the support offered is effective. Liz also described that she had a level of agency over her recovery: *"I needed to sort myself out, I don't think there's anything that could have been done I just needed to sort myself out"*.

Holly spoke about being made aware of the importance of taking responsibility for her mental health whilst in hospital and how it took time for her to be able to understand what this meant:

*"because I remember a nurse in hospital about 3 and a half years ago, she said to me you need to take responsibility and I was like what the fuck are you on about? You've got responsibility over me right now like on a section and all that kind of stuff, you've got responsibility over me I haven't gotta do anything and it wasn't until my last depressive episode I was like oh that's what she means, I have to do stuff too, I'm not just a passive person that takes medication and then leaves hospital"*

*(Holly)*

This concept of responsibility appears to empower young people in allowing them to make choices about the treatment and support that they receive, but also how they choose to make use of these resources. Not all of the participants spoke about responsibility; for example, Annie described in detail how other people were responsible for triggering her self-harm and suicidal ideation. It appeared that she struggled to take responsibility for any role that she may play in the perpetuation of her difficulties; and, appeared to relate her difficulties to the actions of others.

**4.4 It's difficult to think about/ reflect on.** During the interviews, it was clear for the majority of participants that talking about their experiences of attempting to end their life was difficult. There were certain experiences that individuals chose not to talk about and there were times when participants became emotional when reflecting on particular memories. There were also linguistic clues when analysing the transcripts that participants may have been uncomfortable or minimised their experiences, this was done frequently through laughter or long pauses. Liz chose to opt out of sharing certain experiences with the researcher; *"R: and were there many other ways that you tried to end your life apart from overdoses? Liz:. . .yeah. . .R: ok Liz: quite a few, but I don't want to talk about it"*. Belle disclosed how difficult it was to think back to a really difficult time in her life, however explained that on reflection, she can see how she has grown from these experiences:

> *"you know as a kid I was such, I'm a bubbly person, but I lost myself for so long in that time, urm (tearful) sorry. . .looking back at that time, it is, it's really, it's difficult but I'm glad I went through it"*
>
> *(Belle)*

Annie also made it clear how difficult talking about things can be, however she identified that although this is difficult it is also necessary:

> *". . ..you start talking, you're opening old wounds, you're gunna feel like shit but I mean I've been coming to CAMHS since I was in year 7, I've sort of gotten used to the whole things of talking about it and that's what people need to realise, you can't just hold it in, you've gotta talk"*
>
> *(Annie)*

Though participation in this study was not easy for these young people, they all expressed a hope that, through this study, they would be able to help other people who have similar or shared experiences. They stated that reflecting on difficult times is hard yet an important part of recovery from these experiences.

Young people explained how complex their relationships with others can be and how the use of alcohol and substances at times has helped them to cope but has also led to further negatives experiences. They shared difficult memories and reflections and have spoken about the influence of this on their recovery. Some participants were, at the time of interview, still struggling with their mental health whereas others were more settled. Two individuals, with the label of bipolar disorder, described increases in risk in relation to 'mania' and others spoke of overwhelming anxiety and depression. One individual talked about her relationship with voices and how these can also impact on her intentions to end her life.

## Discussion

### Summary of findings

This is the first study to explore the experiences of young people who had attempted suicide in the context of alcohol and/or substance use through in-depth qualitative research. The findings have advanced our knowledge and understanding of how young people make sense of their experience of attempting to end their life, as well as the perceived role alcohol and/or other substance use played within this. The findings of this study demonstrate that the role of alcohol and other substances in relation to suicidal thoughts and behaviour is complex and influences

young people's suicidal experiences in a range of ways. Overall, four superordinate themes were identified: 'the complexity of relationships', 'the double-edged sword of alcohol and substances', 'the straw that broke the camel's back' and 'reflecting on the on-going process of recovery'. The study suggests important clinical implications and reflections on future research in relation to the assessment and management of young people with suicidal experiences.

Findings from our study suggest that there are a number of factors which may influence a young person's decision to end their life. Young people shared their experiences of complex family relationships, interpersonal conflict, different life stressors, the pressure and stress of which gradually builds up until they reach a point when they are no longer able to manage the distress and emotional pain linked to those experiences. In line with the extant literature, this study highlights the role of interpersonal and family conflicts [21, 22] in relation to suicide and suicidal behaviour. The interpersonal theory of suicide [23] posits that suicidal ideation is triggered by two constructs, 'thwarted belonginess' and 'perceived burdensomeness'. However, this theory suggests that the capability to act on these ideas develops through exposure to painful and fearful experiences. Young people within this study described having such experiences in addition to difficulties in feeling connected to others around them which in turn has led some to use alcohol and/or drugs as a way of managing the distress associated with interpersonal difficulties and their struggle to feel connected.

**The complex role of alcohol and substances.**    Although alcohol was the initial focus of the study, many participants also spoke in detail about the use of other substances, such as cannabis and solvents in relation to their suicidal experiences. This highlights the importance of considering multi-substance use, as opposed to a narrow single substance focus, when assessing for suicide risk among young people. Psychological models of addiction, such as Orford's 'Excessive Appetites' [24] model, suggest that there are a range of activities as well as the consumption of alcohol and other drugs, that are deemed 'risky' for individuals who are more predisposed to becoming attached to them. He describes that the processes that underlie these 'appetites' are the same for each behaviour or activity. Therefore, separating alcohol and drug use may not be helpful or necessary. Alcohol and other substance use appeared to be significant methods of coping with interpersonal conflict, overwhelming emotions and life stressors for young people. However, their use was also reported by young people to trigger suicidal feelings, ideation and behaviour. All participants had experienced the use of alcohol and cannabis, three participants had also used other substances such as solvents, MDMA, cocaine and ketamine. Not all participants linked both alcohol and drugs to their suicide attempts. Some participants spoke about cannabis as reducing their levels of anxiety, at least in the short term.

There is limited literature investigating young people's experiences of attempting to end their life. What literature there is appears to highlight the inter and intrapersonal factors which may increase the likelihood of an individual attempting suicide [25]including interpersonal conflict, abuse, previous mental health difficulties, feelings of not belonging, shame, guilt and a struggle to regulate overwhelming emotions. Our study adds value to the extant literature by highlighting the important and complex role that alcohol and/or substances could play within the context of increasing vulnerability to suicide. For some young people in our study, alcohol and/or substances were used as a way of coping with intense pain whereas for others in a more impulsive and unpredictable way leading up to their attempted suicide.

Using drugs and alcohol have been defined as "dysfunctional coping strategies" by some authors and are reported to be more common among individuals who have attempted suicide [26], in particular young people who may be more likely to use alcohol and drugs to self-medicate for difficulties related to mood and anxiety [27]. The findings of the present study demonstrate, through young people's narratives, both the negative consequences but also perceived benefits of the use of alcohol and substances. One young person very clearly stated that she did

not feel that alcohol or cannabis had a negative impact on her mental health, although she did state that she was less able to function the following day, after heavy drinking. This participant spoke about misusing her prescribed medication which was one way in which she had attempted to end her life. For some participants, the use of alcohol and/or substances was a conscious choice i.e. a coping strategy they have learnt from family members which they also adopted as a way of managing distress, anxiety, low mood or hearing voices [28]. However, for others, this was not used in a conscious way to cope with distress but because alcohol was accessible in their home or they drank alcohol or used drugs socially with friends. All participants spoke about how their use of alcohol and substances had changed over time, with many identifying that they had chosen to monitor or reduce the amount they were using due to the repercussions for their mental health.

Young people also spoke about an increase in understanding of both their mental health experiences but also the impact of alcohol and substance use. This increased awareness appeared to facilitate the development of alternative coping strategies. They all reflected on their ongoing journey towards 'recovery', they discussed taking responsibility for their recovery and acknowledged that although their experiences were difficult to reflect on, this reflexion was beneficial.

The findings from this study shed light on the complex and complicated processes underlying a young person's alcohol and/or substance use in the context of attempting to end their life. Acknowledging and embracing this complexity is important in identifying, assessing and managing alcohol and substance use as key suicide risk among young people.

## Strengths and limitations

This is the first study to explore the sense that young people make of attempted suicide in the context of alcohol and/or substance use. There were a variety of experiences discussed by participants, who had been given a range of different diagnoses. Some participants had used a number of different substances alongside alcohol and some had just used cannabis in addition, some had multiple experiences of attempting to end their life and others had just one. This variability can be seen as both a strength and a limitation of the study. Whilst there were many differences between participants, there were still a number of shared experiences and sensemakings across the group, which provide a strength to the results captured.

The method of analysis, IPA, allowed rich, detailed interviews with the participants which were then analysed thoroughly. This helped provide an in-depth account of young people's experiences. A potential limitation of the present study is the representativeness of the sample. All participants identified as White British and 6 out of the 7 participants identified as female. Therefore, the findings of the study are limited in the generalisability to those that are not from a White British background. The experiences and sense making discussed by the participants are likely to have been impacted by their own characteristics and intersectionality which may be different to those from different cultures, races and ethnicities. Gender differences also need to be investigated, whilst the one male participant discussed many shared experiences as his female counterparts, he also spoke about experiences which were different.

## Clinical implications

This study has important implications for suicide prevention in young people. The young people in this study spoke about the accessibility and normalisation of alcohol use and the impact that this has on their suicidal ideation and attempts. A recent systematic review of international literature [29] found that by changing policies related to accessibility and cost of alcohol there was a decrease in suicide rates. The results highlight the importance of including assessment and formulation of alcohol and substance use when working with young people with

suicidal ideation. As explained in the Results, the functions and process' relating to alcohol consumption and suicide attempts are complex. Practitioners should aim to not only identify whether or not a young person is drinking alcohol, but also aim to understand the function of this use to enable a more holistic and individualised overview of their risk of suicide and intervention needs. Interventions should include interpersonal dimensions, for example family and systemic therapies. This increase of familial understanding promotes increased resilience within the family and systems which has been linked with improved outcomes for young people [30]. The Department of Health [31] (DoH) and Health Education England [32] (HEE) acknowledge the importance of supporting the family, as a whole, in improving young people's mental health. National Institute for Health and Care Excellence (NICE) guidelines [33] suggest the use of family therapy for young people with identified alcohol use difficulties. Participants spoke positively about their experiences of DBT and the coping strategies that they had learnt through engaging in this therapy. Supporting patients to build their self-worth and helping instil hope that these thoughts will pass and that their circumstances can be different may provide an opportunity to mitigate some of the risk posed to the self at these times [34].

## Future research

Although a small-scale study, it is one of the first to explore in depth young people's experiences of attempted suicide within the context of alcohol and substance use. The results have broadened our understanding of how young people make sense of these experiences. Further research needs to sample a more diverse population, focusing on those from black and minority ethnic groups and males. Due to the high numbers of completed suicides within young males, this is a priority for researchers. We know that suicide rates are higher among young men of Black African and Caribbean origin, compared to their White peers [35]. Thus, exploring the potential role alcohol and/or substances play in the suicidal behaviour of Black African and Caribbean men is important in order to provide a tailored and evidence-based approach to the assessment and management of suicide risk in this population.

## Conclusion

The young people that took part in this study reported complex and multifaceted experiences of attempting to end their life and their subsequent recovery from this. Their accounts suggested that the use of alcohol and other drugs played an important role, yet the ways in which this occurred varied across participants and seems more complex than the literature has illustrated to date. Participants described use of alcohol and other substances as a way to escape unwanted and overwhelming emotions, deal with interpersonal conflict and traumatic memories, and manage comorbid mental health experiences such as voices and mania. However, alcohol and other substance use as a style of coping in turn may contribute to an increase in suicidal ideation and actions. The findings from this study, although preliminary and in need of further investigation, inform a greater understanding of the many functions of alcohol and substance use and their relationship to suicidal ideation and behaviour in young people. Results can also inform how these processes can be explored by services in order to gain a more comprehensive assessment of support needs which would lead to tailored suicide prevention interventions.

## Supporting information

**S1 Appendix. Consolidated criteria for reporting qualitative studies (COREQ): 32-item checklist.**
(DOCX)

**S2 Appendix. Recruiter information sheet.**
(DOCX)

**S3 Appendix. Recruitment poster.**
(DOCX)

**S4 Appendix. Consent for researcher contact form.**
(DOCX)

**S5 Appendix. Participant information sheet.**
(DOCX)

**S6 Appendix. Additional quotes.**
(DOCX)

## Author Contributions

**Conceptualization:** Alex Copello, Maria Michail.

**Data curation:** Rebecca Guest.

**Formal analysis:** Rebecca Guest.

**Project administration:** Rebecca Guest.

**Supervision:** Alex Copello, Maria Michail.

**Writing – original draft:** Rebecca Guest.

**Writing – review & editing:** Alex Copello, Maria Michail.

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
