## [Decision Letter · Decision Letter 0]

25 Jan 2021

PONE-D-20-36043

A qualitative exploration of young people’s experiences of attempted suicide in the context of alcohol and substance use.

PLOS ONE

Dear Dr. Guest,

Thank you for submitting your manuscript to PLOS ONE. After careful consideration, we feel that it has merit but does not fully meet PLOS ONE’s publication criteria as it currently stands. Therefore, we invite you to submit a revised version of the manuscript that addresses the points raised during the review process.

We look forward to receiving your revised manuscript.

Kind regards,

Vincenzo De Luca

Academic Editor

PLOS ONE

Journal Requirements:

Reviewers' comments:

Reviewer's Responses to Questions

**Comments to the Author**

1. Is the manuscript technically sound, and do the data support the conclusions?

Reviewer #1: Partly

2. Has the statistical analysis been performed appropriately and rigorously? 

Reviewer #1: Yes

3. Have the authors made all data underlying the findings in their manuscript fully available?

Reviewer #1: No

4. Is the manuscript presented in an intelligible fashion and written in standard English?

Reviewer #1: Yes

5. Review Comments to the Author

Reviewer #1: Thank you for the opportunity to review this manuscript. This is an important topic for a vulnerable population that requires an in-depth analysis to inform appropriate, future interventions. My specific comments are appended below.

ABSTRACT

• It is somewhat difficult to follow the themes placed in quotations. I would suggest that the authors either find new ways to describe and/or organize their key themes

• This manuscript would be greatly improved if the authors detailed the percentage of participants who endorsed a given theme

• The abstract highlights barriers. Were facilitators also explored?

INTRODUCTION

• The authors highlight risk factors of suicide. Are these limited to completed suicide or also relevant for ideation? Can a sentence or two also highlight protective factors?

• It is not clear how the authors differentiate ‘alcohol’ and ‘substance use’

• Please add appropriate references for statistics: e.g., “Young people (over 18) who engage in binge drinking have been found to be 2.6 times more likely to attempt suicide”

METHODS

• Inclusion/exclusion criteria: need to be better justified.

o The authors list that having a ‘self-defined problematic use of alcohol (and substances’ was required as inclusion criteira; however, it is known that adolescents may not recognize that they have issues with substances, especially alcohol. Can the authors justify why this criteria was required?

o The authors exclude those who have made a suicide attempt in the past 6 months. Why was this the case when the authors previously argue that it is important to examine those with alcohol issues to inform suicide interventions?

• Can the authors justify why they used a sample size of 7 and not less or more? Was saturation of specific themes considered for the listed sample size?

• Interviews being conducted by the local mental health team may introduce bias. Independent and trained research personnel are typically the gold standard for qualitative research. A rationale for why this interviewee was selected for the current study should be detailed and potentially recognized as a limitation in the Discussion section.

• Who conducted the debrief session that followed the qualitative interview?

• It is not clear who conducted the analysis and whether inter-rater reliability was reached prior to completing all analyses

RESULTS

• Table 2 is informative however it is not clear what the third column (“Participants that contributed”). These results could be bolstered by highlighting the frequency and percentage of participants who endorsed a given theme

• The theme “Keeping safe from others’ does not seem to appropriately reflect the longer text. It sounds like all participants felt misunderstood by people around them

• It may worthwhile to organize the themes accordingly to barriers and facilitators. This organization is particularly informative to interventionists.

• It is not clear whether the data was analyzed for stratified groups e.g., younger vs. older youth; male vs. female

DISCUSSION

• Additional limitations including the small sample size should be noted. It is not clear whether these results are generalizable to contexts outside of the UK or to other, non-white groups

6. PLOS authors have the option to publish the peer review history of their article (what does this mean?). If published, this will include your full peer review and any attached files.

Reviewer #1: No

---

## [Author Response · Author response to Decision Letter 0]

28 May 2021

Thank you for your comments. Please see attached document titled response to reviewers.

---

## [Editor Report · Decision Letter 1]

19 Aug 2021

A qualitative exploration of young people’s experiences of attempted suicide in the context of alcohol and substance use.

PONE-D-20-36043R1

Dear Dr. Guest,

We’re pleased to inform you that your manuscript has been judged scientifically suitable for publication and will be formally accepted for publication once it meets all outstanding technical requirements.

Kind regards,

José J. López-Goñi

Academic Editor

PLOS ONE
---

## [Editor Report · Acceptance letter]

23 Aug 2021

PONE-D-20-36043R1 

*A qualitative exploration of young people’s experiences of attempted suicide in the context of alcohol and substance use.*

Dear Dr. Guest:

I'm pleased to inform you that your manuscript has been deemed suitable for publication in PLOS ONE. Congratulations! Your manuscript is now with our production department. 

Kind regards, 

on behalf of

Dr. José J. López-Goñi 

Academic Editor

PLOS ONE